# Solving Social Problems in Aging Rural Japanese Communities: The Development and Sustainability of the Osekkai Conference as a Social Prescribing during the COVID-19 Pandemic

**DOI:** 10.3390/ijerph182211849

**Published:** 2021-11-12

**Authors:** Yumi Naito, Ryuichi Ohta, Chiaki Sano

**Affiliations:** 1Department of Community Medicine Management, Faculty of Medicine, Shimane University, 89-1 Enya Cho, Izumo 693-8501, Japan; somayum38@gmail.com (Y.N.); sanochi@med.shimane-u.ac.jp (C.S.); 2Community Care, Unnan City Hospital, 96-1 Iida, Daito-Cho, Unnan 699-1221, Japan

**Keywords:** social prescribing, rural, Osekkai, Osekkai conference, COVID-19, Japan

## Abstract

Social prescribing is critical during the coronavirus disease 2019 (COVID-19) pandemic. Social prescribing refers to non-medical interventions related to culture and traditions; they are increasingly being proposed to address wider determinants of health as well as help patients improve health behaviors and manage their conditions effectively. Traditional and cultural behaviors in the community can be utilized for effective social prescribing. Due to the pandemic, social participation opportunities have decreased, resulting in the absence of Osekkai, a traditional Japanese behavior. A driver of Osekkai is the Osekkai conference; it is the model through which Osekkai is implemented in communities to bring resources and individuals together to address community needs. This research aims to clarify the Osekkai conference’s development process and how it can solve social problems, thereby leading to the creation of sustainable communities. We conducted semi-structured interviews with Osekkai conferences’ participants and organizers during the COVID-19 pandemic. We used thematic analysis to achieve the research aims. A total of 12 participants were interviewed. Five themes were developed from the thematic analysis: driver of the Osekkai conference’s development, trajectory of continuity, chain of Osekkai, changes in communities and participants, and sustainability of the conferences. This study showed how reviving traditional behaviors helps face initial difficulties. It described these increasing traditional behaviors in terms of social prescribing that changes the community’s and citizens’ social capital. Reviving traditional behaviors created new challenges and solutions during the pandemic.

## 1. Introduction

An increase in the number of aging individuals and their health conditions has become more complicated [1]. Health determinants include various social problems, such as isolation [2], low social capital [3], and little social support [4], with a low quality of social relationships. These factors can deteriorate society’s healthcare. Social problems can be contextual because each community has specific issues related to its cultural, socioeconomic, and geographical issues [5].

Each community must address its social problems through discussions. These should include not only laypersons, but also various stakeholders, such as the government, medical care providers, and social professionals [6,7]. Effective collaboration can formulate pragmatic solutions to social problems, which can in turn result in better health care [8]. Local governments and community workers can drive this collaboration. Thus, social prescribing is a process of addressing aging and isolated societies. In social prescribing, non-medical interventions related to culture and traditions are proposed to address wider determinants of health and to help patients improve health behaviors and manage their conditions effectively [9]. Social prescriptions worldwide have the potential to establish communities’ sustainability by supporting various people’s social lives [9].

Citizen-driven community discussions regarding community development related to culture and traditions can be productive for the creation and maintenance of continuous problem-solving systems in communities [10]. Community development refers to community building, which can assist citizens to live comfortably and actively [11]. Social prescribing may be beneficial for effective community development [9]. In social prescribing, community workers can connect people who need help with social resources [12,13]. Here, citizens’ empowerment is vital for discussion [14]. Moreover, citizen-driven community discussions can be critical for aging societies. Since there are various forms of discussion, the clarification of the ways and processes of solving social problems is vital for the continuation of discussions on effective social problem solving [15]. However, there is a lack of evidence on the process and concrete effectiveness of the participants.

Osekkai, a traditional Japanese behavior in the community, can be utilized as a form of social prescribing. With Osekkai, people create a safe and comfortable community through collaboration. In turn, this situation can increase social participation among organizations and individuals who implement community programs. Osekkai is an unconscious behavior within Japanese communities, which are based on collectivist values [16]. In their communities, Japanese people are culturally educated to perform Osekkai, which they consider as acting for the betterment and comfort of others. However, aging and multiculturalism reduce Osekkai, thus harming the relationships within these communities [17]. Due to the coronavirus disease 2019 (COVID-19) pandemic, opportunities for social participation have decreased, resulting in the dwindling of Osekkai within communities. In addition, Osekkai can be viewed as an invasion of privacy and thus avoided in other communities. Nonetheless, as insiders, community workers and members of the community can identify various difficulties in their communities, which can drive effective Osekkai. Revitalizing this practice can help rebuild social connections within rural communities [16]. According to previous research, Osekkai has been associated with higher social participation in rural people during the COVID-19 pandemic [18]. Therefore, Osekkai should be encouraged in rural communities to ensure sustainability.

The Osekkai conference, a driver of Osekkai, is organized by nurses in rural communities. Through this conference, community nurses create opportunities for citizens who face difficulties in their communities. The nurses attempt to resolve these difficulties by motivating citizens to discuss community issues and respect culture and traditions [16]. They facilitate conference discussions and connect citizens with social resources related to culture and traditions, similar to social prescribing [9]. People struggle to solve social problems related to their health conditions in Japanese rural areas, where aging affects people’s lives [19]. The Osekkai conferences can motivate rural communities to discuss solutions by involving various stakeholders, such as medical care professionals and local government staff [20]. Therefore, the conference may empower rural communities to discuss their social problems. This study’s research questions are as follows: “How did the Osekkai conference begin”, “How did it progress and overcome challenges”, and “What are the present challenges regarding the continuity and progression of the Osekkai conference”. In this study, we sought to clarify the conference’s development process, find solutions, and present challenges that are critical for practical community development to improve human relationships worldwide. Indeed, showing that reviving traditional Japanese behaviors is effective can trigger the revival of traditional behaviors in other cultures, thus fostering communities with better human connections. Finally, this study explores how the conference can solve social problems to create sustainable communities.

## 2. Methods

This study performed qualitative research involving Osekkai conferences’ participants and organizers to clarify the conferences’ development process and how they can solve social problems, leading to the creation of sustainable communities.

### 2.1. Setting

Unnan City, located in the southeast of the Shimane Prefecture, is one of the most rural areas in Japan. In 2020, Unnan’s total population was 37,638 (18,145 males and 19,492 females), with 39% being over 65 years old. This proportion of people above 65 years old is expected to reach 50% by 2025 [21]. The situation in the city is typical of Japanese rural settings. There are 30 autonomous community organizations in Unnan City. Each community organization has various functions for managing social issues, such as social isolation, accessibility to medical care, and succession of traditional activities. Each district has at least one autonomous community organization [21].

### 2.2. The Osekkai Conference

Emergent situations require voluntary activities in communities where human resources are limited and isolated people are unable to seek help. In Japan, both formal and informal rural care providers establish flexible meetings called Osekkai conferences. These conferences allow people to come together and discuss community problems. Conferences currently occur in small groups to avoid the spread of COVID-19 in rural communities [16]. At the Osekkai conference, rural residents including people whose families and friends have social problems, community health workers, and healthcare professionals present their community’s problems and social issues. By collaborating with the Unnan City Hall, welfare organizations, and hospitals, community organizers invite various professionals with in-depth understanding of the city’s social problems to connect participants with various resources. Thus, the conferences involve people with varying professional backgrounds, including medicine, law, public health, rehabilitation, architecture, community development, and transportation. Conference attendees collaborate to solve community problems by sharing experiences and suggesting solutions to similar problems. Finally, after establishing an Osekkai plan, it is implemented in each rural community using the suggested resources from the conference. Then, the results of the plan’s provisions are shared in the succeeding Osekkai conferences. Osekkai plans can be revised to improve the quality of care through continuous reflection and discussions. These routine discussions can foster new and effective collaborations among providers of different resources (Figure 1). The subject conference of this study began in September 2020, and mainly took place in Kisuki, Unnan City. Osekkai conferences were held at a private house in Kisuki, Unnan City Japan, which was familiar for the participants, thus ensuring a safe place to share the private information of the participants as well as community problems. By the end of March 2021, 19 conferences were held, and 302 participants participated in the conferences, of which 74.1% were from Unnan City [18].

### 2.3. The Participants

The participants were rural citizens who participated in the Osekkai conferences wherein people discussed social problems regarding their communities’ difficult living conditions. The participants were selected through purposive sampling respecting their roles in Osekkai conferences. The Osekkai conference had five types of activities: consulting on community difficulties with conference staff, planning, performing, supporting, and accepting Osekkai. The participants gathered at the community center where the conference was held on the conference day. Purposive sampling was employed to recruit interview participants by respecting their role in the Osekkai conferences. The participants included citizens who were helped through Osekkai, citizens who supported the provision of Osekkai, and citizens who organized the Osekkai conferences.

### 2.4. Data Collection

We conducted semi-structured interviews that lasted for about 60 min each. The interviews were conducted through a two-way videoconferencing software, Zoom, from 1 September 2020 to 31 December 2020. These interviews were facilitated by the first and second authors. Audio was recorded during the interview, and their contents were transcribed verbatim after. The three authors analyzed the transcriptions based on thematic analysis. The interview guide included four main questions: “How did the Osekkai conference start?”, “How did the quality of Osekkai conferences improve?”, “What do you think of the benefits of Osekkai conferences to their participants and organizers?” and “How do you think this conference can continue for the creation of sustainable communities?”.

### 2.5. Analysis

Thematic analysis was used to identify the Osekkai conference’s development and its process of solving social problems, thus leading to the creation of sustainable communities [22]. The thematic analysis comprised six steps: familiarization with data, generating initial codes, searching for themes, revising themes, defining and naming themes, and producing the report. The first and second authors carefully studied the interview transcriptions. The first author subsequently coded the content and developed codebooks based on his repeated reading. The second author independently coded the content. The two authors discussed the coding based on the codebook. In this process, the authors inducted, merged, deleted, and refined concepts and themes by comparing their research materials and coding [22]. Qualitative data from transcribed interviews were coded using NVivo software to elicit core themes. Discussions regarding the data and coding continued until a mutual agreement was reached and all possible concepts or themes had already been exhausted, leading to theoretical saturation [22]. Finally, all authors discussed and agreed on the themes and concepts. Furthermore, the results were translated from Japanese to English.

### 2.6. Ethical Consideration

The participants were informed prior to attending the research and before the start of the interviews that the data collected would be used only for research purposes. Additionally, the participants were apprised of the research aims, how the data would be disclosed, how their personal information would remain confidential, and that they could stop the interview at any time if desired. The participants provided written consent prior to the study. Finally, this study was approved by the Rural City Hospital Clinical Ethics Committee (approval code: 20190021).

## 3. Results

Twelve participants were interviewed, and 33.3% (four participants) were male. Specifically, four participants were citizens who received aid through the Osekkai (recipient), four who supported the provision of Osekkai (supporter), and four who organized the Osekkai conferences, consulted on community difficulties with the conference staff, or planned and performed Osekkai (provider). Five themes and fourteen concepts were developed using thematic analysis (Table 1).

### 3.1. Driver of the Development of the Osekkai Conference

For the first question, “How did the Osekkai conference start?”, the following finding was clarified. The Osekkai conference began owing to the limitations of present systems in rural communities that were administered by local governments. Collaboration among various organizations is required for problem solving in rural areas, especially when dealing with rural communities’ complex problems.

#### 3.1.1. Limitation of Governments

Originally, local governments functioned well in addressing usual problems previously occurring in communities based on the established systems for solutions. However, rural communities’ present conditions are quite different from those in the past. Rapidly aging societies and complicated community structures have changed communities’ constitution. As such, citizens’ lives were encroached upon. Some participants stated:
*“The present communities can vary. Older people have different issues from the past, such as isolation and various medical problems. Young generations also have problems in terms of continuity of living in rural areas, such as those concerning working conditions and relationships with others. The government has systems to solve old types of problems. However, they cannot solve the present problems because they need speed in coping with the present social changes.”*(Participant 1, provider.)
*“To decide on anything, governments may need time to discuss among their various departments and stakeholders. Time is important in creating sophisticated systems. However, various people suffer from present social problems. We should act more quickly than that. In this COVID-19 pandemic, many people are isolated and should be quickly helped through flexible systems.”*(Participant 4, provider.)

Rapidly changing societies driven by the COVID-19 pandemic impinged present governmental systems in rural communities. Therefore, new systems were expected to quickly solve rural community problems in a manner fitting for each issue.

#### 3.1.2. Needs for Relationship among Various Organizations

Rural social problems included various aspects that crossed the borders of one specialty. Some people need help from delivery services and shop clerks to go purchase their needs. Various people and organizations must collaborate to help in these people’s lives. Some participants stated:
*“For solving various social problems, all aspects of the problems can be approached by one organization. Present social problems can be caused by various factors not only from persons but also from social and medical problems. For example, handicapped and disabled people’s shopping can be difficult in rural areas. They need a system to transmit them to a shopping place. Moreover, they need help systems there. They need comprehensive support from various organizations.”*(Participant 2, provider.)
*“We need more collaboration among different organizations to solve the present social problems. With respect to collaboration, we need a place to discuss social problems among people coming from different groups.”*(Participant 5, supporter.)

The present rural problems could not be solved simply. Various collaborations among different organizations are required to effectively address such problems. Thus, establishing a collaboration among these organizations was vital. These needs have led to the Osekkai conference’s establishment.

### 3.2. Trajectory of Continuity

For the second question, “How did the quality of Osekkai conferences improve?”, the following findings were clarified. To launch the Osekkai conferences, community nurses and organizers gathered participants and managed their motivations for the conferences. Moreover, they constructed a friendly atmosphere for Osekkai and advanced organizers through continuous reflection. Additionally, the organizers induced Osekkai to sustain the continuity of the Osekkai conferences.

#### 3.2.1. Construction of an Atmosphere Friendly to Osekkai

Osekkai was considered to be awkward or redundant prior to launching the Osekkai conferences because of individualism’s spread in Japan. Osekkai conferences’ organizers attempted to motivate rural citizens to construct their plans for Osekkai in communities. They did this through dialogs with rural citizens and local governments. One participant stated:
*“Osekkai might not have good images, but relatively bad images for rural citizens. Therefore, we first had to dispel the bad images. To mitigate the bad images, we considered that rural citizens should have the time to construct plans for Osekkai. This will provide them with an opportunity to consider Osekkai’s goodness. We tried to discuss the difficulties leading to the construction of Osekkai with rural citizens and local governments. Some of them were motivated and empowered. While the image of Osekkai may not be changed drastically, some citizens may feel more than what they originally felt toward Osekkai in the past. Thus, they may be able to consider Osekkai without hesitation.”*(Participant 12, supporter.)

Initially, Osekkai’s image was not acceptable to the communities. However, organizers’ efforts and support from motivated citizens and local government officers contributed to a positive outlook toward Osekkai.

#### 3.2.2. Continuous Reflection on Organizing

The organizers had to revise the manner in which the conferences were held to further develop the Osekkai conferences. These organizers likewise had to reflect on their motivations for the conferences. At first, the conference’s organization was challenging because the participants were not used to considering Osekkai in depth. The organizers had to make presentations explaining Osekkai. Moreover, they had to continuously provide meanings of its presence in each conference. Furthermore, the organizers attempted to create many opportunities to entice the participants to talk with various participants. It must be noted that Osekkai needs various collaborations to be successful; therefore, such a move was undertaken by the organizers to establish various connections among the participants. Some participants stated:
*“The beginning of this conference was difficult to manage. All things have their firsts. Even though Osekkai had a bad image as redundant thing or invasion of privacy, concrete Osekkai should be explained to the participants. The participants understood the meaning of Osekkai to some extent. However, when they tried to consider the concrete Osekkai, they could not comprehend the plan of Osekkai at first. Therefore, we had to take steps toward Osekkai or support the plans of Osekkai.”*(Participant 9, organizer.)
*“Successful Osekkai needs various collaboration. We intended to create various dialogs among the participants. At first, the dialogs’ management was difficult because many participants went out after the whole discussion. We reflected on the organization of the conferences and revised the contents inducing various dialogs after the conferences. We did this by talking to the participants so that they would participate in dialogs after the conferences.”*(Participant 8, organizer.)

The organizers attempted to manage the participants’ learning about Osekkai. They likewise encouraged dialogue among the participants to create an effective collaboration among the organizers and participants. The organizers needed continuous reflection for the conferences’ management.

#### 3.2.3. Induction of Osekkai by Organizers

The beginning of the Osekkai conference required organizers’ strong facilitation. Initially, few people suggested plans for Osekkai. Thus, the organizers had to motivate various community people to consider their community issues and difficulties. People cloistered themselves and were reluctant to go out during the COVID-19 pandemic. The organizers experienced difficulties in promoting Osekkai conferences given their community conditions.


*“People’s initial hesitation toward Osekkai can be attributed to Japanese culture. Initially, I had to instruct community people to consider community issues to extract community difficulties. The COVID-19 pandemic is impinging on our lives, so it was difficult to collect people in the Osekkai conferences. As such, finding people motivated to do Osekkai was vital for the continuity of the conferences. Motivated people stimulate other participants to consider Osekkai in depth.”*
(Participant 5, supporter.)

For Osekkai’s induction, continuous participant facilitation and finding motivated people were vital to progress the conferences. Additionally, this was crucial to motivate other participants.

### 3.3. Chain of Osekkai

For the second and third questions, “how did the quality of Osekkai conferences improve?” and “what are the benefits of Osekkai conferences to their participants and organizers?”, the following findings were clarified. Through continuous efforts to hold Osekkai conferences, Osekkai’s perception was transformed from a negative image to that of a positive one—that which encourages behaviors. Each Osekkai was effectively performed by constructing and reappraising relationships and utilizing the original social capital. Furthermore, motivated citizens began their own constructed Osekkai in their communities by receiving support from Osekkai conferences’ organizers.

#### 3.3.1. Dispelling the Negative Image of Osekkai

Osekkai’s negative image prevented rural citizens from considering and performing it. Rural citizens considered the performance of Osekkai as an invasion of privacy. An increasing number of citizens changed their perception of Osekkai through these conferences and the promotion of its positive perceptions among communities. These citizens, then, started embarking on Osekkai. One participant stated:
*“At first, I felt that the provision of Osekkai itself seemed strange because it was awkward. However, by participating in the conference, my image of Osekkai changed slightly, from a thing that should be avoided to a thing that can be done for our communities.”*(Participant 6.)

Another participant stated:
*“It was hard to continue holding an Osekkai conference. However, I saw the gradual changes in the attitudes of the participants of the conferences. They became more motivated to consider Osekkai as compared to how I saw them at the beginning.”*(Participant 9, provider.)

#### 3.3.2. Social Capital Driving Osekkai

Each participant in the conference had various connections with others. Osekkai’s recipients also had various human connections in their communities as social capital. The connections became intertwined and dense through these conferences, thereby making the performance of each Osekkai effective. They were driven by the social capital of the participants involved in Osekkai. One participant stated:
*“In planning and performing Osekkai, we realized that there were various connections among people in communities. By connecting them, Osekkai could be more fruitful. Moreover, others as social capitals could become active by being involved in Osekkai.”*(Participant 3, supporter.)

The Osekkai conferences provide the participants with the opportunity to reconsider the human relationship in their communities. Further, they activated social capital in the provision of Osekkai.

#### 3.3.3. Spontaneous Appearance of Osekkai

The initial induction of Osekkai by the organizers motivated people in the communities. Some of them began to consider their community problems in depth. This led to a new plan for Osekkai without organizers’ inductions and provisions for concrete Osekkai with the organizers’ support. One participant stated:
*“I felt doing Osekkai is meaningful by participating in this conference. Through Osekkai, the providers and recipients of Osekkai became motivated and activated in their lives. Their difficulties in the communities are mitigated. Therefore, I started to consider the plan for Osekkai based on the issues in my communities.”*(Participant 10, receiver.)

Motivated participants’ and recipients’ lives became active by participating in this conference and being involved in concrete Osekkai. This induced the spontaneous appearance of Osekkai in communities.

### 3.4. Changes in Communities and Participants

For the third question, “What are the benefits of Osekkai conferences to their participants and organizers?” the following findings were clarified. Through multiple Osekkai conferences, the connections in the communities were reconstructed. In particular, this was done by intertwining and condensing social capital. This, in turn, created new interactions among different generations. The increase in community variety created new and varied problems in their communities. The Osekkai conferences were then held in each community through an outreach program because the problems in communities were context-specific.

#### 3.4.1. Reconstruction of Social Capital

Through the provision of Osekkai, new relationships were reconstructed based on their previous relationships. Each Osekkai involved various people who previously knew each other, but had not met for a long time. Furthermore, some people could help others who had previously helped them in Osekkai. This contributed to stronger connections among them. One participant stated:
*“As part of Osekkai, I could get the help from the customers who used my shop. I became inactive because of my disease. I was disappointed. They supported me by transferring me to a car and from the car to the shopping mall. I appreciate them, and I am happy to be involved in the Osekkai conferences. With these conferences, I have regained previous relationships.”*(Participant 4, receiver.)

Another participant stated:
*“I participated in a provision of Osekkai. In the Osekkai, I met a previous friend who made an effort to help others. By knowing his efforts, I was motivated and collaborated with him.”*(Participant 9, provider.)

The opportunities for involvement in Osekkai enabled the lost relationships in communities to be reconstructed. Further, these motivated the participants to live in communities.

#### 3.4.2. Interaction among Different Generations

Some Osekkai were involved in various generations, from younger to older people. Different generations collaborated with each other to perform Osekkai in the conference. This contributed to the generation of new connections among them. There was a lack of frequent interactions between generations as rural areas began aging. One participant stated:
*“Recently, I have not talked with children and young generations. I did not imagine that I would collaborate with them. Through Osekkai conferences and Osekkai, I could now talk with them and provide Osekkai. I realized that I could connect with the younger generations even now.”*(Participant 2, provider.)

Another participant stated:
*“The Osekkai conference can construct new relationships in communities that could be formed in the past when citizens frequently gathered and had dialogs in societies.”*(Participant 1, provider.)

The Osekkai conferences determined that the participants must reconstruct new relationships that were precious in the past communities. They must help each other in sustaining communities’ functions.

#### 3.4.3. Advanced Complexity of Communities

Osekkai conferences’ expansion connected participants with different priorities, leading to complex issues in communities. There were misunderstandings and conflicts among the participants. The organizers had to facilitate the conferences’ dynamics to promote Osekkai. They tried to respect each participant’s statements and guide the discussion to develop productive ideas. One participant stated:
*“Now, various people participate in conferences and share their ideas. In communities, there are people with different characteristics and mental disorders. Their perspectives can be different, and their statements can stimulate other participants. In the internet-based platform of the Osekkai conference, which aimed to increase interaction among the participants, their posting could disturb some discussions. Therefore, organizers have to facilitate their statements.”*(Participant 3, supporter.)

The participants positively perceived the situation. In this era of the COVID-19 pandemic, unprecedented things may happen in the future. The organizers and participants felt that community complexity should be encouraged. Moreover, different community members should collaborate and solve their issues from different perspectives. One participant stated:
*“We will be exposed to various difficulties in the future, including the COVID-19 pandemic. We should adjust to various situations and use them to change people’s lives. As such a variety of communities can be important. We can consider various things from different perspectives. Communities are becoming increasingly inclusive.”*(Participant 7, receiver.)

The communities of the Osekkai conferences became more diverse during the COVID-19 pandemic. Consequently, they became inclusive for minorities through the Osekkai provisions.

#### 3.4.4. Outreaching of the Conference

The locality of the issues of the communities forced the Osekkai conferences to become diverse. This pertains not only to the members but also the place where the conferences are held. The COVID-19 pandemic negatively affected the holding of Osekkai conferences. The Osekkai conference was held as a form of an outreach program to mitigate infection risk and focus on community conditions. One participant stated:
*“The Osekkai conferences had to adjust their focus on each particular community. This is because of the increase in the numbers and diversity of the issues suggested from the participants. The organizers decided that the Osekkai conferences should be held in each community with the organizers and local participants. The outreach was effective in driving Osekkai in each community.”*(Participant 8, provider.)

Another participant stated the following regarding the COVID-19 pandemic:
*“The large-scale conference became dangerous, and the participants feared that the infection is spreading. Therefore, the conferences’ small-scale outreach in each community is reasonable for infection control.”*(Participant 2, provider.)

### 3.5. Sustainability of the Conferences

For the fourth question, “how can this conference continue to create sustainable communities?”, the following findings were clarified. For the Osekkai conferences’ progression, a structure of continuity was required. The conferences provided an opportunity to collaborate with public organizations that were supported by the central and local governments. Through discussions among the organizers, a collaboration between the Osekkai conferences and public post offices was arranged, and the post office was designated as a venue for the conference. Furthermore, conferences were diffused to other communities by assigning headquarters in different communities, thereby ensuring the sustainability of the conferences.

#### 3.5.1. Collaboration with Public Officials

For the conferences’ continuity, the organizers realized that they should collaborate with organizations with long histories. Furthermore, these organizations must be supported by the central and local governments. One of the organizations was a post office supported by the Japanese government. Post offices are located in each community and the citizens of the community are familiar with such post offices. The organizers viewed the collaboration with the post offices as a sustainable choice that would ensure the mobilization of participants. Furthermore, this would help them organize the conferences effectively by identifying various social problems in communities due to diverse participation. One participant stated:
*“To consider the continuity of the conference, we had to collaborate with more sustainable organizations. Our choice was a post office. People in communities use post offices regularly. Therefore, officers know the situations of communities in depth. We considered that the collaboration could find issued routes in communities, leading to sustainable approaches in resolving community issues.”*(Participant 11, receiver.)

Another participant stated:
*“For the post offices, we had various anxieties toward community’s issues. However, we did not have the methods to effectively approach the issues. Through collaboration with the Osekkai conferences, we can acquire a way to effectively approach these issues.”*(Participant 5, supporter.)

The collaboration between the Osekkai conferences and post offices in rural areas was beneficial for solving community issues. In fact, it was one of the solutions for the continuity of the Osekkai conferences.

#### 3.5.2. Diffusion of the Conference to Other Communities

The increase in the number of participants was vital for Osekkai conferences’ sustainability. At first, the conferences were held in a limited area of the city. After the progression of the Osekkai conferences’ organization, the conferences began to be held in other areas in the city. This took place through the collaboration with the local government and communities. By expanding the areas, the number of people who became interested and participated in the conferences increased. One participant stated:
*“A lot of people in the city got interested in our conferences. This reputation was formed by participants who performed or supported various Osekkai communities. They talked about this conference with their friends in other communities, which motivated various people to participate in these conferences. This led to the conference being held in other communities.”*(Participant 4, receiver.)

Another participant stated:
*“The effective collaboration with the local government and community organizers was productive. When they understood the importance of the conferences, they actively promoted this conference in the city and in each community. For sustainability, both the local government and community organizers motivated participants and organizations to support our activities.”*(Participant 2, provider.)

## 4. Discussion

This study demonstrates the processes of the Osekkai conferences, and its initial difficulties and challenges. Further, it shows the possibility of improvement of Osekkai’s quality, which could lead to changes in social capital among citizens and communities. As such, this triggered new challenges and led to solutions during the COVID-19 pandemic. Osekkai conferences faced various difficulties in the beginning. These include bad images of Osekkai as something redundant or an invasion of privacy and varied characteristics of citizens. The conferences’ continuity and efforts of Osekkai conference organizers in revising the provision of the conferences made it possible to form effective relationships among participants. This, in turn, could lead to new social capital, as confirmed by previous research on Osekkai conferences, showing an increase in the number of interactions with others [18]. New and already constructed social capital was mixed through Osekkai’s provision. The mixture of social capital led to effective Osekkai and induced spontaneous Osekkai from motivated participants. Osekkai conferences’ expansion resulted in a variety of participants taking part in the conferences. As such, conflicts in the communities arose. However, as diversity was considered to be respected for inclusive communities, the organizers managed to deal with the conflicts. Moreover, they attempted to make the participants understand the community’s diversity. For Osekkai conferences’ sustainability, the organization collaborated with various stakeholders, such as public organizations, local governments, and community organizations.

Well-established systems and relationships collapsed during the COVID-19 pandemic. As this study shows, governmental administrations’ limitations were clarified in rural settings. Governments’ decision making and provision of help, commonly through speaking, are formative and not flexible. Governmental help should be aligned to each condition because each community has different problems based on social and cultural conditions in the COVID-19 pandemic [23,24]. However, the government usually formulates one size-fits-all plans. As this study shows, rural areas with various communities needed personalized help, driving citizen-centered organizations [25,26]. Furthermore, communities need different kinds of help from various organizations to solve problems. Collaboration among various organizations is mandatory to solve complicated problems in communities. Social problems worldwide have become diverse. Moreover, the COVID-19 pandemic drove them to become more diverse and critical because of the impingement of human relationships [27,28]. Diverse problems’ solutions, complicated by COVID-19, should be approached by citizen-oriented organizations based on sociocultural traits such as social prescribing.

The continuity of citizen-driven activities requires participants’ motivation and problem orientation among communities. In this study, the organizers started by making the atmosphere Osekkai-friendly. Furthermore, the way of the Osekkai conferences was revised creatively through reflections among participants and communication in communities. Moreover, participants’ ideas were respected, possibly inducing spontaneous Osekkai in communities. There has been a limitation of human interaction during the COVID-19 pandemic, thereby restricting most social activities. As such, social capital among citizens is reduced [29,30]. Social capital reduction may impinge on the health conditions of people in communities, especially among the older and the isolated [31]. The Osekkai conferences could connect these people with social resources in communities. Thus, the difficulties they experience in their communities are mitigated. Positive experiences in communities motivate them to carry out Osekkai toward others to improve their community conditions. Since social capital can be invisible, citizen-driven activities can find new social capital for each citizen. These activities can likewise bridge the citizens with already-preserved social capital. This leads to higher interwoven social capital contributing to health conditions [32,33]. Citizen-driven activities in the community should be encouraged with the help of governments during the present COVID-19 pandemic.

Traditional activities can increase social capital during the COVID-19 pandemic. They can likewise foster sustainability through collaboration with government-driven organizations and outreach based on participants’ reputations. This study shows that the changing atmosphere in communities regarding the traditional activity of Osekkai and Osekkai conferences could increase the interaction of community members across different generations with respect to infection control measures. This increase in interactions may rebuild social capital within the communities. The interactions of different groups within communities and countries can trigger various problems due to differences in religion, traditions, and group traits. These differences may negatively affect the collaboration among different groups [34,35,36]. The Osekkai conferences had the same problems regarding community conflicts. However, these problems were overcome by engaging in community issue-oriented activities and respecting said differences. People can collaborate to overcome their differences in emergent situations. They may likewise work together to solve the same difficulties. Organizing people’s collaboration with respect to their differences can help solve community problems effectively, thus increasing social capital in different groups [37,38]. Furthermore, the sustainability of citizen-driven traditional activities can be secured by collaborating with government-driven organizations [39]. This collaboration can improve citizens’ accessibility to Osekkai conferences and financial support. The Osekkai conferences’ involvement in government-driven organizations ensures continuity and reliability to citizens, as governments are stable. During the COVID-19 pandemic, the sustainability of citizen-driven traditional activities should be ensured by collaborating with various organizations, as this can lead to the development of sustainable communities.

Traditional behaviors in other cultures and contexts can effectively improve social capital within communities. Traditional behaviors that increase interactions in communities can change people’s relationships and should be facilitated by community workers [7,37]. These behaviors should be considered in the framework of social prescribing [12,13]. Understanding traditional behaviors as social prescribing can contribute to community solutions by using original community resources, such as people’s relationships and social and healthcare resources [34]. However, during the COVID-19 pandemic, people could not use various resources, including social and healthcare resources, because of the possibility of infection [40]. This trend can increase comprehensiveness in communities, but each community must solve its problems by itself. In these situations, traditional behaviors should be revived in various contexts, and individuals should help each other to ensure the sustainability of the community.

This study had certain limitations. First, we focused on rural citizen-driven activities in Japan. Rural areas in other countries may be dealing with similar conditions. There may likewise be fewer human interactions and people in less congested areas. This study’s findings may be transferrable to other rural contexts not only to improve social activities but also to enhance social capital in rural contexts. Nonetheless, this study’s findings should be generalized to other rural contexts with caution. Second, this study may have limited reliability. This study was performed during the COVID-19 pandemic, and participants had already experienced various pandemic stages. As such, the perception of the participants in the Osekkai conferences could have changed over time. In this study, we attempted to examine the changes that have occurred in Osekkai conferences during the COVID-19 pandemic. The changes in the conferences can also be generalized to rural contexts by adding quantitative data collection and analysis, thus improving their transferability to other contexts. Third, this study only used one-on-one interviews; we did not quantitatively assess Osekkai conferences’ effectiveness in rural communities. Therefore, future research should quantitatively explore traditional activities, the sustainability in rural and other contexts, and the changes in community conditions in rural contexts by measuring the changes in the number of conference participants and their social capital and isolation status.

## 5. Conclusions

This study investigated the process of managing initial difficulties and challenges in the Osekkai conferences. Furthermore, it shows potential ways for improving the quality of Osekkai to increase the social capital of communities and their members. The created relationships in the communities, in turn, have created new challenges and solutions during the COVID-19 pandemic. For the activities’ sustainability, future studies should quantitatively investigate the effectiveness of these activities as social prescribing in relation to social capital and isolation. Furthermore, this study described how reviving traditional behaviors can help face difficulties and challenges at the onset. It presented these traditional behaviors as examples of social prescribing, leading to changes in the community’s and citizens’ social capital. These changes created new challenges and solutions during the COVID-19 pandemic.

## Figures and Tables

**Figure 1 ijerph-18-11849-f001:**
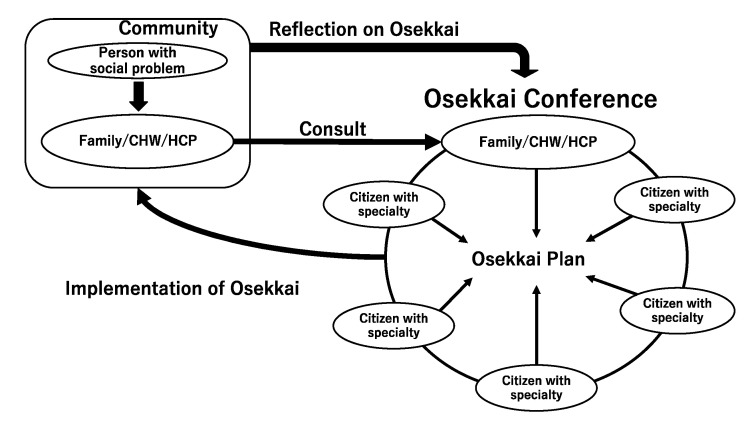
Conceptual framework of Osekkai Conference.

**Table 1 ijerph-18-11849-t001:** The results of the thematic analysis.

Theme	Concept
Driver of the Osekkai conference development	Limitation of governments
	Needs for the relationships among various organizations
Trajectory of continuity	Construction of a friendly atmosphere toward Osekkai
	Continuous reflection on organizing
	Induction of Osekkai by organizers
Chain of Osekkai	Dispelling the negative image of Osekkai
	Social capital driving Osekkai
	Spontaneous appearance of Osekkai
Changes in communities and participants	Reconstruction of social capital
	Interaction among different generations
	Advanced complexity of communities
	Outreaching of the conference
Sustainability of the conferences	Collaboration with public officials
	Diffusion of the conference to other communities

## Data Availability

The datasets used and/or analyzed during the current study available from the corresponding author on reasonable request.

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
