# Peer review of "Solving Social Problems in Aging Rural Japanese Communities: The Development and Sustainability of the Osekkai Conference as a Social Prescribing during the COVID-19 Pandemic"

_ijerph, 2021, doi:10.3390/ijerph182211849_

Round 1

Reviewer 1 Report

Thank you for your submission and for sharing this interesting approach to developing community collaboration and expanding community capacity. For persons who are unfamiliar with Osekkai, could you provide some additional details on what practices differentiate Osekkai from other community development approaches? Also is there an underlying philosophical framework undergirding this practice?  What was the theoretical framework guiding your study?  Could you provide a clearer definition of social prescribing? How does it differ from collaborating to solve problems and making referrals?  You mention that Osekkai is an unconscious behavior. Can you speak more to how it is unconscious? How did the facilitators at these conferences connect people to resources?  This approach sounds like you held a community forum with an inter-professional team representing different segments of health care and social service industries. Can you speak a bit more concretely about how you successfully got participants to engage in small group dialogue and motivated them to attend.  What types of negative images did participants have of Osekkai? You also mentioned that organizers had to "control the statements" of participants. (Lines 368-369). This language is a bit heavy handed. It does not sound collaborative. Can you explain a bit more about how you guided the discussion as a facilitator? Finally, how did you connect people with resources? How formalized was the referral process? Thank you for clarifying these questions. Overall, the manuscript conveyed some important general principles but needs to more clearly articulate concrete applications so the reader has a better idea of how the information might be transferrable to a different context.

Author Response

For persons who are unfamiliar with Osekkai, could you provide some additional details on what practices differentiate Osekkai from other community development approaches?

Response:

Thank you for highlighting this. We agree with this comment. Based on your suggestion, we have revised the introduction section by including the importance of Osekkai and the reasons for its negative perception in other communities (Page 2, 3).

Also is there an underlying philosophical framework undergirding this practice?

Response:

Thank you for highlighting this. We agree with this comment. Based on your suggestion, we have added information on the values of collectivism in Japanese culture in the introduction section.

What was the theoretical framework guiding your study?

Response:

Thank you for highlighting this. We agree with this comment. Based on your suggestion, we have included the theoretical framework of the Ossekai conferences on Page 3.

Could you provide a clearer definition of social prescribing?

Response:

Thank you for highlighting this. We agree with this comment. Based on your suggestion, we have added the definition of social prescribing in the second paragraph of the introduction section.

How does it differ from collaborating to solve problems and making referrals?

Response:

Thank you for highlighting this. We agree with this comment. Based on your suggestion, we have revised the explanation of Osekkai conferences in the introduction and methods sections by showing that Osekkai conferences add traditional Japanese behaviors to collaboration in communities.

You mention that Osekkai is an unconscious behavior. Can you speak more to how it is unconscious?

Response:

Thank you for highlighting this. We agree with this comment. Based on your suggestion, we have included the unconscious behaviors in the introduction section.

How did the facilitators at these conferences connect people to resources?

Response:

Thank you for highlighting this. We agree with this comment. Based on your suggestion, we have revised the explanation of Osekkai conferences by adding organizers’ functions that are aimed at connecting people to resources (Page 3, 4).

This approach sounds like you held a community forum with an inter-professional team representing different segments of health care and social service industries. Can you speak a bit more concretely about how you successfully got participants to engage in small group dialogue and motivated them to attend. 

Response:

Thank you for highlighting this. We agree with this comment. Based on your suggestion, we have revised the explanation of Osekkai’s conference and its collaboration with other facilities in the city.

What types of negative images did participants have of Osekkai?

Response:

Thank you for highlighting this. We agree with this comment. Based on your suggestion, we have revised the results section by adding the negative perceptions of Osekkai as an invasion of privacy.

You also mentioned that organizers had to "control the statements" of participants. (Lines 368-369). This language is a bit heavy handed. It does not sound collaborative. Can you explain a bit more about how you guided the discussion as a facilitator?

Response:

Thank you for highlighting this. We agree with this comment. Based on your suggestion, we have revised the suggested part from control to facilitate.

Finally, how did you connect people with resources? How formalized was the referral process? Thank you for clarifying these questions. Overall, the manuscript conveyed some important general principles but needs to more clearly articulate concrete applications so the reader has a better idea of how the information might be transferrable to a different context.

Response:

Thank you for highlighting this. We agree with this comment. Based on your suggestion, we have revised the explanation of the Osekkai conference, by highlighting the specialty of this interventions as well as its application to other settings.

Reviewer 2 Report

Dear authors 

I would like to congratulate you on your excellent work here. I only have a minor concern about the used methodology, qualitative analysis has some blunted edges that are necessary to be further explained. 

As the paper is attached, maybe it could be a guide to explain something more about thematic analysis.https://pubmed.ncbi.nlm.nih.gov/31367394/ 

In the same line, a more elaborate table, instead of table 1, could be desiderable. 

Ossekai Conference is a great tool and resource to help the public health system and could be an exportable and adaptable tool. I recommend the author to point and indicate by a list of bullet points how others could adapt this system. 

Best wishes 

Author Response

I would like to congratulate you on your excellent work here. I only have a minor concern about the used methodology, qualitative analysis has some blunted edges that are necessary to be further explained. 

As the paper is attached, maybe it could be a guide to explain something more about thematic analysis.https://pubmed.ncbi.nlm.nih.gov/31367394/ 

Response:

Thank you for highlighting this. We agree with this comment. Based on your suggestion, we have added the six steps of thematic analysis.

In the same line, a more elaborate table, instead of table 1, could be desiderable. 

Response:

Thank you for highlighting this. We agree with this comment. As the results section is long, we have created Table 1 to help readers comprehend the results of this study.

Ossekai Conference is a great tool and resource to help the public health system and could be an exportable and adaptable tool. I recommend the author to point and indicate by a list of bullet points how others could adapt this system.

Response:

Thank you for highlighting this. We agree with this comment. Based on your suggestion, we have added the concrete method of the conference and the inclusion of professionals and participants.

Reviewer 3 Report

The presented paper deals with the social changes of the COVID-19 pandemic in Japan. This is per se very interesting and an important field of research, although the limitations of the peculiarities of Japanese society must also be taken into account.

The introduction is very well written and introduces the reader to the topics position and is very well supported with literature.
Maybe it would be nice to present the research question/hypothesis to the reader at the end.
The methodology is well described. It should be added here according to which criteria were the subjects included in the study?
In addition, the chosen mode of evaluation seems very subjective and should be critically questioned by the authors again!

The compilation of the results is not entirely clear. Generalized statements are made and one or two quotes from interviewees are added. the reader is missing the entire picture of the interviewees. The authors should find a model here according to which the answers are standardized and presented in their entirety. Thus the impression remains for the reader that the evaluation took place purposefully. 

The evaluation is detailed and the results found are discussed with current literature.

Author Response

The introduction is very well written and introduces the reader to the topics position and is very well supported with literature.

Response:

Thank you for highlighting this. We have drafted a more elaborate introduction section, with keen attention to the definition of Osekkai and social prescribing.

Maybe it would be nice to present the research question/hypothesis to the reader at the end.
The methodology is well described. It should be added here according to which criteria were the subjects included in the study?

Response:

Thank you for highlighting this. We agree with this comment. Based on your suggestion, we have added the research questions, selection criteria, and the use of purposive sampling (Page 3, 4)

.

In addition, the chosen mode of evaluation seems very subjective and should be critically questioned by the authors again!

Response:

Thank you for highlighting this. We agree with this comment. Based on your suggestion, we have revised the description of the thematic analysis by adding the methods of thematic and qualitative analyses based on the NVIVO software.

The compilation of the results is not entirely clear. Generalized statements are made and one or two quotes from interviewees are added. the reader is missing the entire picture of the interviewees. The authors should find a model here according to which the answers are standardized and presented in their entirety. Thus the impression remains for the reader that the evaluation took place purposefully. 

Response:

Thank you for highlighting this. We agree with this comment. Based on your suggestion, we have revised the description of each of the themes by adding more information to explain them.

The evaluation is detailed and the results found are discussed with current literature.

Response:

Thank you for highlighting this. We have added the meanings of unconscious behavior and negative perceptions of Osekkai in the results and discussion sections.

Reviewer 4 Report

Thank you for submitting this interesting paper for review.

The concept of Osekkai is interesting and has the potential to be useful outside of Japan. However, the average reader of IJERPH would not be familiar with Osekkai so more information needs to be presented in the introduction.

ABSTRACT/INTRODUCTION: The abstract and the introduction do not appear to go together. If the purpose of the paper is to show how Osekkai would assist in social prescribing for older, isolated adults then social prescribing needs to be defined in the abstract too. The Introduction should provide more background on Osekkai, how long has it been a tradition, when and why did it fell out of favor, who convenes the conferences, who has the authority to implement decisions, who is responsible for collecting results, etc.

The Osekkai process would be easier to understand if it could be represented in a figure.

METHODS: you interviewed 12 people, 4 recipients, 4 supporters, and 4 providers. This is out of how many of those who participated in each group? How did you choose the 4, why choose only 4, and how do you know they were representative of the group they represented? No quantitative data were collected; only qualitative date were discussed. How do you know you got representative data?

Line 59: you state that Osekkai can be utilized for effective social prescribing but isn't that what you want your data to show? This may be a premature statement.

Line 118: how many participants? How many are rural; how were they chosen?

Line 127: what is a distant communication application?

Line 130: are you referring to 2 authors or to 3?

Line 156: how do the members of the 3 groups (recipient, supporter and provider chosen and is there overlap between these groups?

Several of your word choices should be reconsidered. For example:

Line 31: recommend changing its to their.

Line 31: Recommend changing have to has

Line 42: Recommend changing solving to addressing

Line 51 Consider changing facilitate to assist

Line 69: Consider changiing driven to offered

Line 70: change insure to promote

Line 170 change rapid to rapidly

Line 172: how were their lives impinged upon?

Line 196: change likewise to also

Line 212: was recently (how recently was this?)

Line 225: why  or how was Osekkai's image not acceptable?

Line 272: how did you measure the change from negative to positive?

Line 319 you discuss the connections between generations. Do you have quantitative data measuring those reconnections?

Line 356: not sure what you mean by provided. Perhaps determined?

Line 361: what do you mean by personalities? Do you mean priorities?

Line 367 you discuss the social media form of the Osekkai conference. What is that?

Line 369: control? do you mean carefully craft?

Line 404: this is the first time you mention post offices. The idea of meeting in a safe, familiar place should be discussed well before line 404 (and perhaps it could be noted in the figure describing Osekkai conferences.

Line 405: how does being is different places impact the sustainability.

Line 448: you discuss showing the process of the Osekkai conferences. Could you create a figure that shows the process?

Line 449 talks about improvement in Osekkai's quality. What were those quality measures?

Line 456: how did you measure the new social capital?

Line 458: what do you mean by "negated social Osekkai"?

Lines 526: what do you mean by dependability? Do you mean reliability?

Line 536: can you develop a figure to so the process?

How will you measure the sustainability of the Osekkai process?

Author Response

Thank you for submitting this interesting paper for review.

The concept of Osekkai is interesting and has the potential to be useful outside of Japan. However, the average reader of IJERPH would not be familiar with Osekkai so more information needs to be presented in the introduction.

ABSTRACT/INTRODUCTION: The abstract and the introduction do not appear to go together. If the purpose of the paper is to show how Osekkai would assist in social prescribing for older, isolated adults then social prescribing needs to be defined in the abstract too. The Introduction should provide more background on Osekkai, how long has it been a tradition, when and why did it fell out of favor, who convenes the conferences, who has the authority to implement decisions, who is responsible for collecting results, etc.

Response:

Thank you for highlighting this. We agree with this comment. Based on your suggestion, we have added the definition of social prescribing in the abstract, along with a more concrete explanation of Osekkai in the introduction section.

The Osekkai process would be easier to understand if it could be represented in a figure.

Response:

Thank you for highlighting this. We agree with this comment. Based on your suggestion, we have added the figure of the Osekkai conference (Figure 1, Page 4).

METHODS: you interviewed 12 people, 4 recipients, 4 supporters, and 4 providers. This is out of how many of those who participated in each group? How did you choose the 4, why choose only 4, and how do you know they were representative of the group they represented? No quantitative data were collected; only qualitative date were discussed. How do you know you got representative data?

Response:

Thank you for highlighting this. We agree with this comment. Based on your suggestion, we have revised the sampling method. We have used purposive sampling to recruit participants based on their role in the Osekkai conference. We considered the balance of participants’ role in the Osekkai conference while selecting the participants. In the thematic analysis, the researchers discussed the interview contents until theoretical saturation. Thus, the results of this study are credible and reliable.

Line 59: you state that Osekkai can be utilized for effective social prescribing but isn't that what you want your data to show? This may be a premature statement.

Response:

Thank you for highlighting this. We agree with this comment. Based on your suggestion, we have deleted the expression “effective,” as the reviewer suggested that there is no clear evidence of its effectiveness.

Line 118: how many participants? How many are rural; how were they chosen?

Response:

Thank you for highlighting this. We agree with this comment. Based on your suggestion, we have added the number of the participants and their orientation (Page 4)

Line 127: what is a distant communication application?

Response:

Thank you for highlighting this. We agree with this comment. Based on your suggestion, we have used the term “two-way videoconferencing — ZOOM.” (Page 4)

Line 130: are you referring to 2 authors or to 3?

Response:

Thank you for highlighting this. We agree with this comment. Based on your suggestion, we have replaced “the authors” with “three authors.”

Line 156: how do the members of the 3 groups (recipient, supporter and provider chosen and is there overlap between these groups?

Response:

Thank you for highlighting this. We agree with this comment. Based on your suggestion, we have revised the categories of participants based on their roles in the conferences.

Several of your word choices should be reconsidered. For example:

Line 31: recommend changing its to their.

Response:

Thank you for highlighting this. We agree with this comment. Based on your suggestion, we have revised the word.

Line 31: Recommend changing have to has

Response:

Thank you for highlighting this. We agree with this comment. Based on your suggestion, we have revised the term.

Line 42: Recommend changing solving to addressing

Response:

Thank you for highlighting this. We agree with this comment. Based on your suggestion, we have replaced the term.

Line 51 Consider changing facilitate to assist

Response:

Thank you for highlighting this. We agree with this comment. Based on your suggestion, we have revised the term.

Line 69: Consider changiing driven to offered

Response:

Thank you for highlighting this. We agree with this comment. Based on your suggestion, we have revised the term..

Line 70: change insure to promote

Response:

Thank you for highlighting this. We agree with this comment. Based on your suggestion, we have revised the term.

Line 170 change rapid to rapidly

Response:

Thank you for highlighting this. We agree with this comment. Based on your suggestion, we have revised the term.

Line 172: how were their lives impinged upon?

Response:

Thank you for highlighting this. We agree with this comment. Based on your suggestion, we have revised the term.

Line 196: change likewise to also

Response:

Thank you for highlighting this. We agree with this comment. Based on your suggestion, we have revised the term.

Line 212: was recently (how recently was this?)

Response:

Thank you for highlighting this. We agree with this comment. Based on your suggestion, we have revised the expression to “prior to launching the Osekkai conferences.”

Line 225: why  or how was Osekkai's image not acceptable?

Response:

Thank you for pointing this out. We agree with this comment. Based on your suggestion, we have added the explanation of the unacceptability of Osekkai (Page 8).

Line 272: how did you measure the change from negative to positive?

Response:

Thank you for highlighting this. We agree with this comment. Based on your suggestion, we did not conduct any quantitative measurements to assess the changes in the image/perceptions of Osekkai. This result should be validated using quantitative research; this has been included in the limitations section (Page 14, at the end of the discussion section).

Line 319 you discuss the connections between generations. Do you have quantitative data measuring those reconnections?

Response:

Thank you for highlighting this. We agree with this comment. Based on your suggestion, we did not conduct quantitative measurements on the connection between generations. This result should be validated using quantitative research; this has been mentioned in the limitations section (Page 14).

Line 356: not sure what you mean by provided. Perhaps determined?

Response:

Thank you for highlighting this. We agree with this comment. Based on your suggestion, we have revised the term.

Line 361: what do you mean by personalities? Do you mean priorities?

Response:

Thank you for highlighting this. We agree with this comment. Based on your suggestion, we have revised the term.

Line 367 you discuss the social media form of the Osekkai conference. What is that?

Response:

Thank you for highlighting this. We agree with this comment. Based on your suggestion, we have revised the term to “internet-based platform.”

Line 369: control? do you mean carefully craft?

Response:

Thank you for highlighting this. We agree with this comment. Based on your suggestion we have deleted the term to avoid confusion.

Line 404: this is the first time you mention post offices. The idea of meeting in a safe, familiar place should be discussed well before line 404 (and perhaps it could be noted in the figure describing Osekkai conferences.

Response:

Thank you for highlighting this. We agree with this comment. Based on your suggestion, we have revised the description of the Osekkai conference in the methods section by adding the explanation of the place.

Line 405: how does being is different places impact the sustainability.

Response:

Thank you for highlighting this. We agree with this comment. Based on your suggestion, we have added the reasons for sustainability (Page 12).

Line 448: you discuss showing the process of the Osekkai conferences. Could you create a figure that shows the process?

Response:

Thank you for highlighting this. We agree with this comment. Based on your suggestion, we have added the figure of the processes of the Osekkai conferences in the methods section (Figure 1).

Line 449 talks about improvement in Osekkai's quality. What were those quality measures?

Response:

Thank you for highlighting this. We agree with this comment. Based on your suggestion, we have revised the phrase to “the potential for improvement in Osekkai’s quality.” Furthermore, we have described the limitations of qualitative research (Page 14).

Line 456: how did you measure the new social capital?

Response:

Thank you for highlighting this. We agree with this comment. Based on your suggestion, we have revised the expression by referring to previous quantitative research on Osekkai conferences.

Line 458: what do you mean by "negated social Osekkai"?

Response:

Thank you for this suggestion. This was an error on our part. Based on your suggestion, we have replaced it with the term “induce.”

Lines 526: what do you mean by dependability? Do you mean reliability?

Response:

Thank you for highlighting this. We agree with this comment. Based on your suggestion, we have replaced it with the term “reliability.”

Line 536: can you develop a figure to so the process?

Response:

Thank you for highlighting this. We agree with this comment. Based on your suggestion, we have added the figure of the processes of Osekkai conferences in the methods section as Figure 1.

How will you measure the sustainability of the Osekkai process?

Response:

Thank you for highlighting this. We agree with this comment. Based on your suggestion, we have revised the limitations section (Page 14) by including the need for quantitative measurements in future studies.

Round 2

Reviewer 1 Report

Thank you for the revisions and clarifications to your manuscript. I have a much better idea now of what social prescribing is and how the Osekkai conference works.  I still believe that it is difficult for readers to glean concrete applications from the manuscript, other than the importance of collaboration and encouraging social connectedness and community empowerment to meet societal needs.  What is missing are more concrete suggestions for how the readership can apply the knowledge gleaned from this study in their own communities of concern.

Author Response

The response to the reviewers’ comments

Thank you very much for reviewing our manuscript and suggesting productive ways to revise it. We have responded to the reviewers’ comments one by one, and the revisions were highlighted in red in the text. Please consider our manuscript for publication.

Reviewer 1

Thank you for the revisions and clarifications to your manuscript. I have a much better idea now of what social prescribing is and how the Osekkai conference works.  I still believe that it is difficult for readers to glean concrete applications from the manuscript, other than the importance of collaboration and encouraging social connectedness and community empowerment to meet societal needs.  What is missing are more concrete suggestions for how the readership can apply the knowledge gleaned from this study in their own communities of concern.

Response:

We would like to thank the reviewer for this insightful comment. We agree with your suggestion and have revised the Abstract, Introduction, and Discussion by including ways in which our results can be transferred to other contexts focusing on culturally traditional behaviors as follows:

Line 27-30

This study showed how reviving traditional behaviors helps face initial difficulties. It described these increasing traditional behaviors in terms of social prescribing that changes the community’s and citizens’ social capital. These changes created new challenges and solutions during the pandemic.

Line 79 to 98

The Osekkai conference, a driver of Osekkai, is organized by nurses in rural communities. Through this conference, community nurses create opportunities for citizens who face difficulties in their communities. The nurses attempt to resolve these difficulties by motivating citizens to discuss community issues and respect culture and traditions [16]. They facilitate conference discussions and connect citizens with social resources related to culture and traditions, similar to social prescribing [9]. People struggle to solve social problems related to their health conditions in Japanese rural areas, where aging affects people's lives [19]. The Osekkai conferences can motivate rural communities to discuss solutions by involving various stakeholders, such as medical care professionals and local government staff [20]. Therefore, the conference may empower rural communities to discuss their social problems.

This study’s research questions are as follows: “How did the Osekkai conference begin,” “How did it progress and overcome challenges,” and “What are the present challenges regarding the continuity and progression of the Osekkai conference.” In this study, we sought to clarify the conference’s development process, find solutions, and present challenges that are critical for practical community development to improve human relationships worldwide. Indeed, showing that reviving traditional Japanese behaviors is effective can trigger the revival of traditional behaviors in other cultures, thus fostering communities with better human connections. Finally, this study explores how the conference can solve social problems to create sustainable communities.

Line 577 to 607

Traditional behaviors in other cultures and contexts can effectively improve social capital in communities. Traditional behaviors that increase interactions in communities can change people’s relationships and should be facilitated by community workers [7,37]. These behaviors should be considered in the framework of social prescribing [12,13]. Understanding traditional behaviors as social prescribing can contribute to community solutions by using original community resources, such as people’s relationships and social and healthcare resources [34]. However, during the COVID-19 pandemic, people could not use various resources, including social and healthcare resources, because of the possibility of infection [40]. This trend can increase comprehensiveness in communities, but each community must solve its problems by itself. In these situations, traditional behaviors should be revived in various contexts, and individuals should help each other to ensure the sustainability of the community.

This study had certain limitations. First, we focused on rural citizen-driven activities in Japan. Rural areas in other countries may be dealing with similar conditions. There may likewise be fewer human interactions and people in less congested areas. This study’s findings may be transferrable to other rural contexts not only to improve social activities but also to enhance social capital in rural contexts. Nonetheless, this study’s findings should be generalized to other rural contexts with caution. Second, this study may have limited reliability.

This study was performed during the COVID-19 pandemic, and participants had already experienced various pandemic stages. As such, the perception of the participants in the Osekkai conferences could have changed over time. In this study, we attempted to examine the changes that have occurred in Osekkai conferences during the COVID-19 pandemic. The changes in the conferences can also be generalized to rural contexts by adding quantitative data collection and analysis, thus improving their transferability to other contexts. Third, this study only used one-on-one interviews; we did not quantitatively assess Osekkai conferences' effectiveness in rural communities. Therefore, future research should quantitatively explore traditional activities, the sustainability in rural and other contexts, and the changes in community conditions in rural contexts by measuring the changes in the number of conference participants and their social capital and isolation status.

Reviewer 4 Report

Excellent revision. Thank you.

Author Response

Excellent revision. Thank you.

Response:

We would like to thank the reviewer for their comment.